# *Mycobacterium leprae* transmission characteristics during the declining stages of leprosy incidence: A systematic review

**Thomas Hambridge**[1]*, **Shri Lak Nanjan Chandran**[1], **Annemieke Geluk**[2], **Paul Saunderson**[3], **Jan Hendrik Richardus**[1]

**1** Department of Public Health, Erasmus MC, University Medical Center Rotterdam, Rotterdam, The Netherlands, **2** Department of Infectious Diseases, Leiden University Medical Center, Leiden, The Netherlands, **3** American Leprosy Missions, Greenville, South Carolina, United States of America

* t.hambridge@erasmusmc.nl

## Abstract

### Background

Leprosy is an infectious disease caused by *Mycobacterium leprae*. As incidence begins to decline, the characteristics of new cases shifts away from those observed in highly endemic areas, revealing potentially important insights into possible ongoing sources of transmission. We aimed to investigate whether transmission is driven mainly by undiagnosed and untreated new leprosy cases in the community, or by incompletely treated or relapsing cases.

### Methodology/Principal findings

A literature search of major electronic databases was conducted in January, 2020 with 134 articles retained out of a total 4318 records identified (PROSPERO ID: CRD42020178923). We presented quantitative data from leprosy case records with supporting evidence describing the decline in incidence across several contexts. BCG vaccination, active case finding, adherence to multidrug therapy and continued surveillance following treatment were the main strategies shared by countries who achieved a substantial reduction in incidence. From 3950 leprosy case records collected across 22 low endemic countries, 48.3% were suspected to be imported, originating from transmission outside of the country. Most cases were multibacillary (64.4%) and regularly confirmed through skin biopsy, with 122 cases of suspected relapse from previous leprosy treatment. Family history was reported in 18.7% of cases, while other suspected sources included travel to high endemic areas and direct contact with armadillos. None of the countries included in the analysis reported a distinct increase in leprosy incidence in recent years.

### Conclusions/Significance

Together with socioeconomic improvement over time, several successful leprosy control programmes have been implemented in recent decades that led to a substantial decline in incidence. Most cases described in these contexts were multibacillary and numerous cases

**Data Availability Statement:** All relevant data are within the manuscript and its Supporting Information files.

**Funding:** The author(s) received no specific funding for this work.

**Competing interests:** The authors have declared that no competing interests exist.

of suspected relapse were reported. Despite these observations, there was no indication that these cases led to a rise in new secondary cases, suggesting that they do not represent a large ongoing source of human-to-human transmission.

## Author summary

Although leprosy remains a public health problem in many parts of the world, several countries have successfully controlled the *Mycobacterium leprae* transmission over recent decades and the case characteristics have been shown to change during periods of declining incidence. However, the potential for certain groups of individuals, such as untreated or relapsed leprosy patients with more severe forms of the disease, to act as sources of ongoing transmission remains unclear. In this systematic review, we aimed to evaluate leprosy case characteristics in low endemic settings, to identify possible sources of transmission and to describe the different control measures implemented. Together with socio-economic improvement over time, the main strategies shared by many of the countries who achieved a substantial reduction in incidence over recent decades included BCG vaccination, active case finding, adherence to MDT and continued surveillance following treatment. By collecting information on cases in these settings, we found that the number of new cases reported remained low with no indication of a rise in new secondary cases, despite a high proportion of multibacillary disease and the presence of persistent cases of suspected relapse. This evidence suggests that such cases do not represent a large ongoing source of human-to-human transmission.

## Introduction

Leprosy is a communicable disease that remains endemic in many areas of the world, although Brazil, India and Indonesia account for approximately 80% of new cases registered globally [1]. It is a chronic infectious disease caused by *Mycobacterium leprae* and (less commonly) *Mycobacterium lepromatosis*, affecting the skin and peripheral nerves of infected individuals [2, 3]. Delay in diagnosis and treatment of leprosy can lead to a wide range of clinical symptoms, often resulting in permanent disfigurement and disability, which in turn can lead to stigma. Transmission pathways of *M. leprae* are not fully understood, although there is evidence of an increased risk of human-to-human transmission for individuals living in close contact with untreated leprosy patients, most likely spread through infectious aerosols [4]. Furthermore, the nine-banded armadillo has been established as another natural host and reservoir of *M. leprae* in the Americas and a potential non-human source of transmission. Although red squirrels in the British Isles were also recently found to harbour the bacterium, no human leprosy cases were ever diagnosed in their surroundings in the past century [5, 6]. In addition, very recently wild chimpanzees have been detected with leprosy, carrying an *M. leprae* strain not known to occur in humans [7].

Since the introduction of multidrug therapy (MDT) in the 1980s, the prevalence of diagnosed leprosy cases has declined by 95%. This decline led the World Health Organization (WHO) to declare leprosy eliminated as a public health problem, defined as a prevalence of less than one leprosy patient per 10,000 population [8]. However, achieving this elimination target was largely attributed to the shortening of treatment duration following the introduction of MDT as well as the clearing of case registers, and did not coincide with a decrease in the

number of new cases detected. This highlights the limitations of using prevalence as an epidemiological indicator of leprosy [9]. In fact, the number of new leprosy cases detected globally only began to decrease after the year 2000, although global annual incidence has more recently plateaued above 200,000 new cases. There were 208,641 new cases reported worldwide in 2019. While the geographical distribution of leprosy is widespread, 95% of all new cases arise from 23 countries which the WHO has declared are global priorities [1].

A number of countries have seen a substantial decline in leprosy incidence through the introduction of various control measures, such as establishing national registries, contact tracing, Bacillus Calmette–Guérin (BCG) vaccination and expanding MDT coverage. Epidemiological patterns of leprosy are typically reported according to certain characteristics, including age, sex, classification, rate of leprosy in children and disability grade [10]. As transmission of *M. leprae* declines in a given population, the profile of new cases has been shown to shift towards older individuals and an increased proportion of multibacillary (MB) cases. This was previously observed in countries such as China, Norway and Portugal [11–13]. One of the main reasons for the shift in subtype is the longer incubation period of MB disease compared to the less severe paucibacillary (PB) form. The same reasoning can also be applied to older populations who in many cases were likely infected several years prior to diagnosis when transmission of *M. leprae* was more frequent or when their immune system was still adequate to suppress bacterial multiplication. Furthermore, the shift towards older age at onset follows a decline in the child leprosy rate, an important measure that indirectly indicates ongoing transmission in the community. On the basis of expert advice, the WHO now uses 'million children' as the denominator when reporting child leprosy rates [1].

It has been previously demonstrated that contacts of individuals with a high bacillary load, for instance MB cases or PB cases with multiple lesions, are at higher risk of being infected with *M. leprae* in endemic areas [14–16]. However, the potential for particular subgroups of individuals, such as untreated or relapsed leprosy patients with multibacillary disease, to act as sources of ongoing transmission remains unclear. The origin of infection can vary considerably depending on the context, comprising of autochthonous cases who contracted the disease within a country or individuals who were likely infected abroad. Investigating the features of leprosy in low endemic settings which experienced a recent decline in incidence may help us gain a better understanding of the overall epidemiology of the disease. Individual case data, case series and epidemiological reports from these contexts can reveal valuable information, such as whether there is evidence of ongoing transmission and what the possible remaining sources of transmission are.

Based on suggestions made by The International Federation of Anti-Leprosy Associations (ILEP) Technical Commission, we conducted a systematic literature review to shed light on the question as to whether transmission of *M. leprae* is driven mainly by undiagnosed and untreated new leprosy cases in the community, or by incompletely treated or relapsing cases. This has been a point of discussion within the leprosy research community and has important policy implications for leprosy control. The aims of our study are three-fold: first, to evaluate the case characteristics during the declining stages of leprosy incidence; second, to identify the possible remaining sources of transmission of cases in low endemic settings; and third, to relate these findings to the different leprosy control measures implemented.

## Methods

### Search strategy

We conducted a systematic literature search of electronic databases in January, 2020 targeting case studies, case series and epidemiological reports in countries that experienced a substantial

**Table 1. Systematic review database and search strategy.**

| Database | Search string |
|---|---|
| Embase | ('leprosy'/exp OR 'Mycobacterium leprae'/de OR 'leprosy control'/de OR (lepros* OR Hansen OR lepra* OR leper*):ab,ti,kw) **AND** ('case report'/de OR 'case study'/de OR 'case finding'/de OR (((case*) NEXT/1 (report* OR stud* OR find* OR series)) OR ((review*) NEAR/3 (literature*))):ab,ti,kw) NOT ((animal/exp OR animal*:de OR nonhuman/de) NOT ('human'/exp)) AND ([ENGLISH]/lim) |
| Medline | (exp "Leprosy"/ OR "Mycobacterium leprae"/ OR (lepros* OR Hansen OR lepra* OR leper*).ab,ti,kf.) AND ("Case Reports"/ OR (((case*) ADJ (report* OR stud* OR find* OR series)) OR ((review*) ADJ3 (literature*))).ab,ti,kf.) NOT (exp animals/ NOT humans/) AND (english).lg |
| Web-of-science | TS = (((lepros* OR Hansen OR lepra* OR leper*)) **AND** ((((case*) NEAR/1 (report* OR stud* OR find* OR series)) OR ((review*) NEAR/2 (literature*)))) NOT ((animal* OR rat OR rats OR mouse OR mice OR murine OR dog OR dogs OR canine OR cat OR cats OR feline OR rabbit OR cow OR cows OR bovine OR rodent* OR sheep OR ovine OR pig OR swine OR porcine OR veterinar* OR chick* OR zebrafish* OR baboon* OR nonhuman* OR primate* OR cattle* OR goose OR geese OR duck OR macaque* OR avian* OR bird* OR fish*) NOT (human* OR patient* OR women OR woman OR men OR man))) AND LA = (English) |
| Cochrane | ((lepros* OR Hansen OR lepra* OR leper*):ab,ti,kw) **AND** ((((case*) NEXT/1 (report* OR stud* OR find* OR series)) OR ((review*) NEAR/3 (literature*))):ab,ti,kw) |
| Google Scholar | Leprosy\|leprae\|lepra case\|cases\|review decline\|declined\|declining |

decline in leprosy incidence through the implementation of control measures. The search string was designed to identify studies that present important case variables. The databases and search terms used are listed below (Table 1). For our search strategy, we adhered to the Preferred Reporting Items for Systematic Reviews and Meta-Analyses (PRISMA) guidelines for our search strategy, an evidence-based minimum set of items for reporting in systematic reviews and meta-analyses [17]. The search strategy used is listed in the PRISMA flowchart (Fig 1). The PRISMA checklist accompanying the PRISMA flow diagram can be found in supplementary data: S1 PRISMA Checklist. This systematic review protocol was registered on PROSPERO under ID CRD42020178923.

## Study selection criteria

The following criteria were used when selecting records retrieved from the literature search:
Inclusion criteria:

- Case reports or case series with individual level data

- Epidemiological reports of aggregated data containing extensive case details

- Studies from countries or regions with less than one new leprosy case detected per 100,000 population

- Descriptions of control measures implemented

Exclusion criteria:

- Summaries of epidemiological trends

- Studies with insufficient case details

- Studies from countries or regions with more than one new leprosy case detected per 100,000 population

- Studies containing data outside of the period of decline

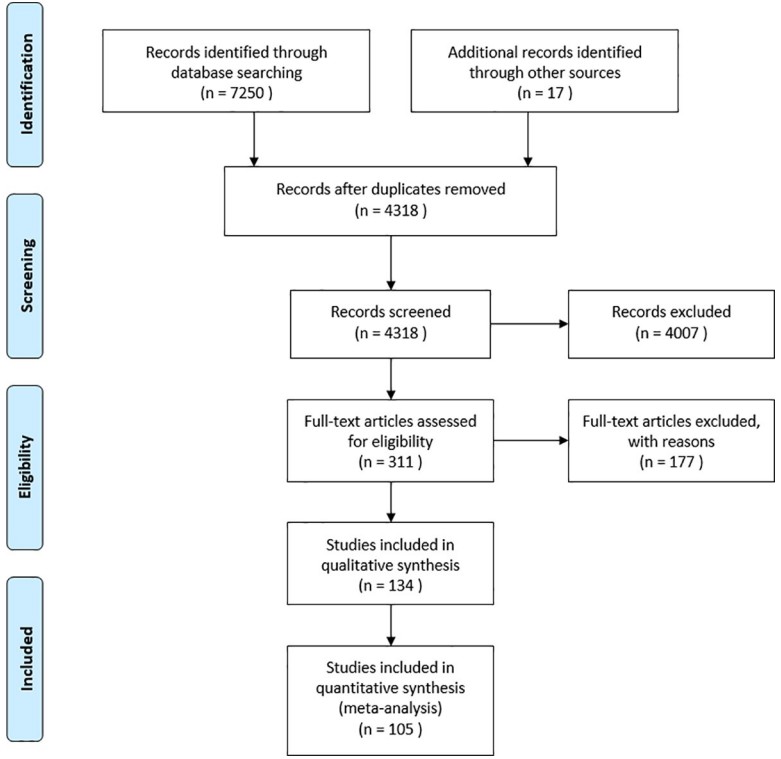

**Fig 1. PRISMA flow diagram.**

In addition to extracting data from peer-reviewed research articles revealed in our search, we also used snowballing (screening of bibliographies for relevant articles) to identify other studies and literature which contained supporting information. Since this review only focused on characteristics of leprosy cases and control measures implemented in low endemic areas, studies in high endemic countries such as India, Brazil and Indonesia were excluded. We defined a low endemic country or region in this study was a case detection rate of less than one new leprosy case detected per 100,000 population in the final year of the timeframe from which they were collected. To collect data during the declining stages of leprosy incidence, we excluded studies that reported findings from earlier time periods of high endemicity from the descriptive analysis.

## Data extraction

One of the authors (TH) developed the search strategy and performed the database search. Two authors (TH and SN) reviewed titles and abstracts for all records retrieved. Once this was complete, the lists were compared and a consensus reached together on which articles were eligible for reading of the full text. Most of the studies consisted of case reports and series which were aggregated by country, while epidemiological summaries were also included where sufficient individual level data was presented. Authors were not blinded to the names of the study authors, journal or institutions (supplementary data: S1 List of Peer-reviewed Studies with Case Data). Data was extracted from the text of relevant articles by two authors (TH and SN) and entered into Microsoft Excel categorised by country.

Variables that were collected from source articles included the number of cases presented, location, year of diagnosis, ethnicity, sex, age, suspected autochthonous or imported, leprosy

subtype (both WHO and Ridley Jopling classification), disability grade, urban or rural, occupation, skin biopsy results, case detection delay (defined as the total time from first signs of leprosy until diagnosis), treatment and duration, surveillance, suspected relapse from previous treatment, suspected reinfection, family history and other suspected source. Autochthonous cases were defined as those suspected to have contracted the disease within the country, while imported cases were considered to have been infected in another leprosy endemic area outside of the country. Additional clinical notes were also recorded which may contain relevant information, such as details on previous exposure to a known leprosy patient or travel to a highly endemic region. The definitions of leprosy subtype followed either the WHO classification system, the Ridley Jopling classification system, or both (Table 2) [18].

## Data analysis and presentation

Since the relevant variables were collected from various types of studies with different methods of data presentation, a descriptive analysis was performed to summarise the findings across several countries. Individual data were imported into Statistical Package for Social Sciences (SPSS) version 26 for analysis. The main quantitative methods available for data presentation were total case count, proportions and means. Due to the heterogeneity in geographical location and timeframe for the data collected, demonstrating an association between case variables and trends in case detection was not an objective of the study. Instead, a qualitative summary was synthesised for each of the 22 countries included in the individual data extraction, as well as an additional two for investigation into their prior control strategies: the Democratic Republic of Congo and Norway. These text summaries outline key findings from the individual level data extraction, epidemiological trends of leprosy and details on the national control measures implemented prior to the decline.

## Results

### Case characteristics

**Summary.** Following a literature search of major electronic databases, we extracted data on a total of 3950 leprosy cases across 22 low endemic countries (Table 3). The ranges for year of diagnosis was different for each country and fell within the period of declining leprosy incidence. Data was collected on several case characteristics available from the source articles (Table 4).

### Demographic information

The average age across all countries was 46.1 years, with the lowest ages observed in cases from Australia, Germany, Libya and Malta, each reporting a mean age under 40 years (Fig 2). The

**Table 2. Diagnosis of leprosy under the WHO and Ridley Jopling classification systems.**

| WHO Classification | Ridley Jopling Classification |
|---|---|
| Paucibacillary (PB) leprosy:<br>1 to 5 skin lesions, without demonstrated presence of bacilli in a slit skin smear | Tuberculoid leprosy (TT)<br>Borderline tuberculoid (BT) |
| Multibacillary (MB) leprosy:<br>Six or more skin lesions; or with nerve involvement (pure neuritis, or any number of skin lesions and neuritis); or with the demonstrated presence of bacilli in a slit skin smear, irrespective of the number of skin lesions | Mid-borderline (BB)<br>Borderline lepromatous (BL)<br>Lepromatous leprosy (LL) |

**Table 3. Sources of individual leprosy case data from 22 low endemic countries.**

| Country | Data sources | Total cases | Year of diagnosis (range) |
|---|---|---|---|
| Australia | 8 | 11 | 1999–2017 |
| Canada | 3 | 186 | 1979–2017 |
| China* | 3 | 785 | 1990–2017 |
| Germany | 6 | 8 | 1994–2016 |
| Iran | 2 | 207 | 1991–2009 |
| Italy | 10 | 27 | 1992–2017 |
| Japan | 9 | 20 | 1990–2017 |
| Libya | 1 | 54 | 1994–1998 |
| Malta | 1 | 136 | 1971–2000 |
| Morocco | 1 | 801 | 2000–2017 |
| Netherlands | 1 | 622 | 1970–1991 |
| New Zealand | 1 | 38 | 2004–2013 |
| Oman | 1 | 77 | 2000–2015 |
| Portugal | 1 | 15 | 1991–2011 |
| Saudi Arabia | 1 | 242 | 2003–2012 |
| South Korea | 1 | 24 | 2009–2013 |
| Spain | 7 | 97 | 1989–2018 |
| Taiwan (Republic of China) | 1 | 81 | 2002–2011 |
| Thailand | 1 | 108 | 1995–2015 |
| United Kingdom | 6 | 11 | 1977–2014 |
| United States | 39 | 304 | 1982–2018 |
| Vietnam | 1 | 96 | 2018 |
| Total | **105** | **3950** | **1970–2018** |

*Shandong province only

mean case detection delay in this study was 31.4 months. A majority of these data points (778/874) came from Chinese leprosy cases, although the mean detection delay was similar in China compared to the rest of the countries with available information (31.7 and 28.8 months, respectively). Almost two-thirds of all cases were male (65.2%), with only Taiwan and Thailand recording a female percentage over 50% (Fig 3).

## Origin of *M. leprae* infection

Information on whether transmission of *M. leprae* was suspected to have originated within the country was described in 2749 of the individual cases included in 18 low endemic countries (Table 5). Overall, slightly more cases of suspected autochthonous transmission were reported (51.7%), although there was a substantial difference observed between countries. From the studies included in this systematic review, there were no reports of foreign-born cases in China. A majority of cases were also suspected to have originated from autochthonous transmission in Iran, Japan, Libya and Oman. On the other hand, several European countries were among those with predominantly foreign-born cases of leprosy recorded, along with Canada, New Zealand and South Korea,. This was particularly true in the Netherlands (96.3%), although many of the cases presented here were recorded several decades earlier. In cases where information on family history was described, 18.7% reported having a family member who was previously diagnosed with leprosy, representing a suspected source of infection.

**Table 4. Overview of case characteristics using combined individual leprosy case data.**

| Characteristic | N | % | Mean |
| --- | --- | --- | --- |
| Age (years) | 1807 | - | 46.1 |
| Case detection delay (months) | | | |
| China | 778 | - | 31.7 |
| Outside of China | 96 | - | 28.8 |
| Both | 874 | - | 31.4 |
| Sex | | | |
| Male | 2529 | 65.2 | - |
| Female | 1351 | 34.8 | - |
| Suspected autochthonous | | | |
| No | 1329 | 48.3 | - |
| Yes | 1420 | 51.7 | - |
| Family history | | | |
| No | 942 | 81.3 | - |
| Yes | 216 | 18.7 | - |
| Subtype (WHO) | | | |
| Paucibacillary (PB) | 1379 | 35.6 | - |
| Multibacillary (MB) | 2497 | 64.4 | - |
| Subtype (Ridley-Jopling) | | | |
| Tuberculoid (TT) | 341 | 23.0 | - |
| Borderline Tuberculoid (BT) | 294 | 19.8 | - |
| Mid-Borderline (BB) | 114 | 7.7 | - |
| Borderline Lepromatous (BL) | 282 | 19.0 | - |
| Lepromatous (LL) | 433 | 29.2 | - |
| Indeterminate (IL) | 21 | 1.4 | - |
| Suspected relapse | | | |
| No | 496 | 80.3 | - |
| Yes | 122 | 19.7 | - |

## Leprosy subtype

The WHO leprosy subtype was much more widely reported and nearly two-thirds (64.4%) of all cases were MB. These findings also varied between countries, with Canada, the Netherlands,

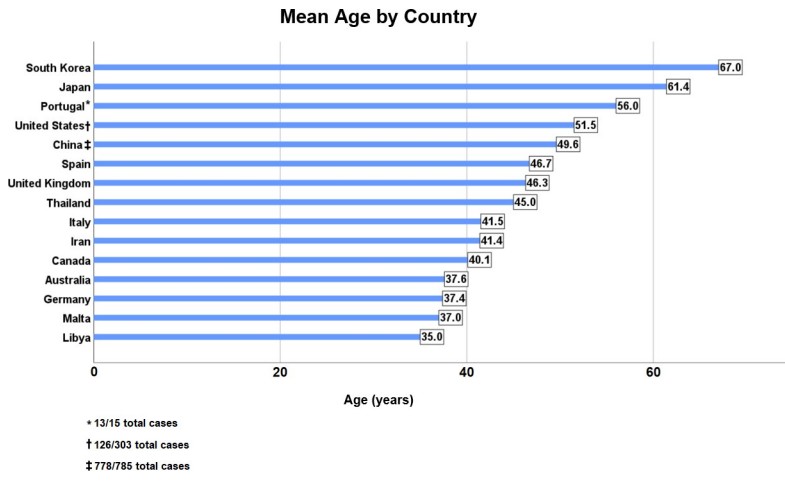

**Mean Age by Country**

* 13/15 total cases
† 126/303 total cases
‡ 778/785 total cases

**Fig 2. Mean age of recorded cases in each country during period of leprosy decline.**

## Sex of Leprosy Cases by Country

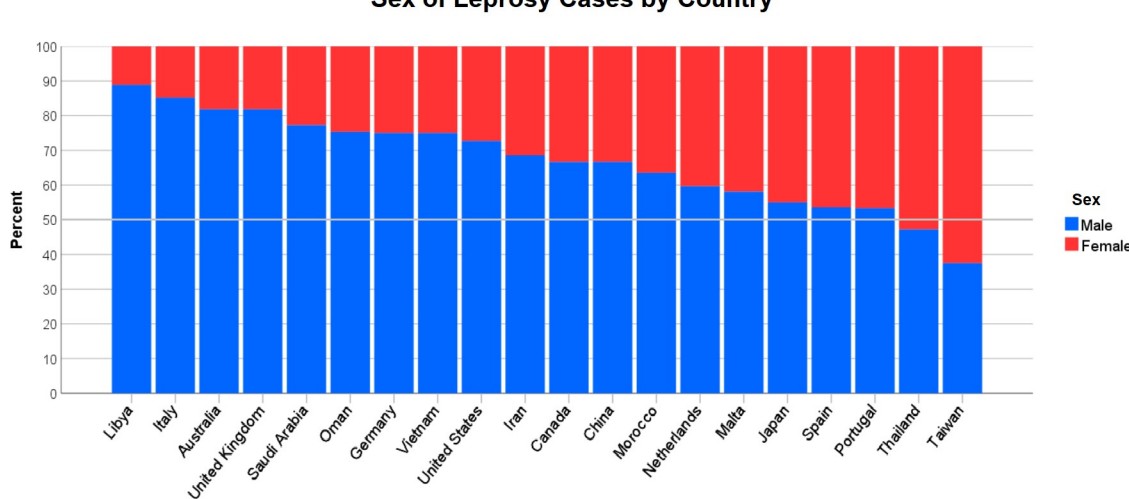

**Fig 3. Percentage of male and female leprosy cases.**

Saudi Arabia and the United Kingdom the only countries reporting more PB cases than MB (Fig 4). In total, 14 countries reported cases classified under the Ridley Jopling system, with LL the most common subtype reported (29.2%), followed by TT (23.0%), BT (19.8%) and BL (19.0%). Iran, Japan, Malta and Portugal had the highest proportions of cases on the lepromatous end of the spectrum (Fig 5).

**Table 5. Suspected imported and autochthonous cases reported.**

| Country | Imported | | Autochthonous | |
|---|---|---|---|---|
| | N | % | N | % |
| Australia | 5 | 55.6 | 4 | 44.4 |
| Canada | 184 | 98.9 | 2 | 1.1 |
| China | 0 | 0.0 | 785 | 100.0 |
| Germany | 7 | 87.5 | 1 | 12.5 |
| Iran | 2 | 1.0 | 205 | 99.0 |
| Italy | 15 | 62.5 | 9 | 37.5 |
| Japan | 4 | 20.0 | 16 | 80.0 |
| Libya | 7 | 13.0 | 47 | 87.0 |
| Netherlands | 554 | 96.3 | 21 | 3.7 |
| New Zealand | 37 | 100.0 | 0 | 0.0 |
| Oman | 24 | 31.2 | 53 | 68.8 |
| Portugal | 8 | 53.3 | 7 | 46.7 |
| Saudi Arabia | 139 | 57.4 | 103 | 42.6 |
| South Korea | 18 | 75.0 | 6 | 25.0 |
| Spain | 56 | 57.7 | 41 | 42.3 |
| Taiwan (Republic of China) | 44 | 54.3 | 37 | 45.7 |
| United Kingdom | 11 | 100.0 | 0 | 0.0 |
| United States | 214 | 72.3 | 82 | 27.7 |
| **Total** | **1329** | **48.3** | **1420** | **51.7** |

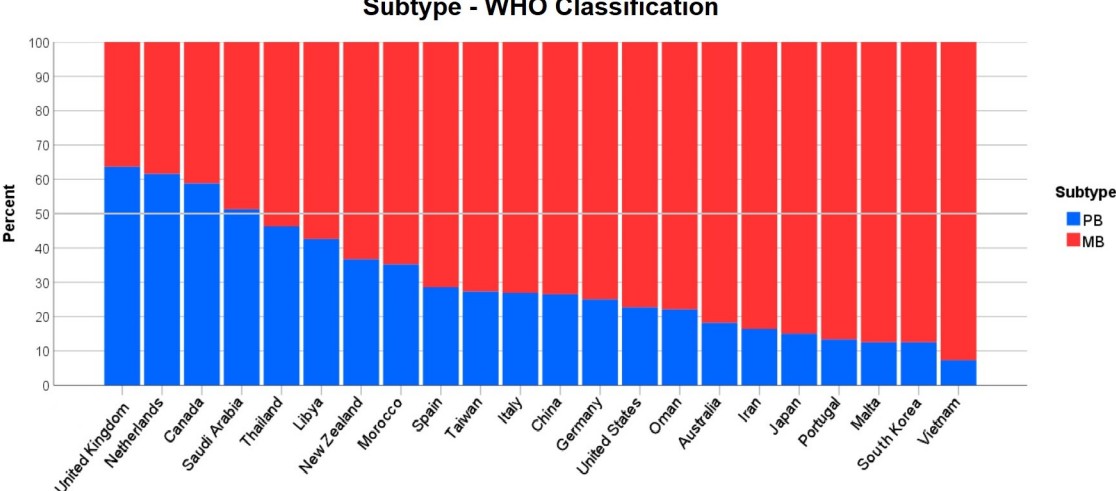

**Fig 4. Leprosy subtype per country using the WHO classification system.**

## Treatment and relapse

Where treatment information was available, most cases received MDT (82.2%) according to prevailing WHO treatment guidelines at the time for PB and MB cases. The remaining cases were treated with different regimens, such as diaminodiphenyl sulfone (DDS) monotherapy or regimens containing ofloxacin and/or minocycline. Cases described in Japan and Malta were mainly treated with alternative regimens. Articles from 12 countries specified whether a case received no prior treatment for leprosy or whether a suspected relapse had occurred following previous treatment, mostly DDS monotherapy, with a total of 122 (19.7%) cases of suspected relapse (Table 6). A total of 69 cases of suspected relapse were reported in Canada, all but one of whom were foreign-born. The other countries which reported a high rate were Iran, Portugal, Spain and the United Kingdom.

## Country descriptions

**Australia.**   There have been very few reported leprosy cases in non-migrants in Australia who had not travelled over recent decades, with local transmission occurring mostly in Aboriginal Australians in the Northern Territory and occasionally Queensland [19]. Leprosy

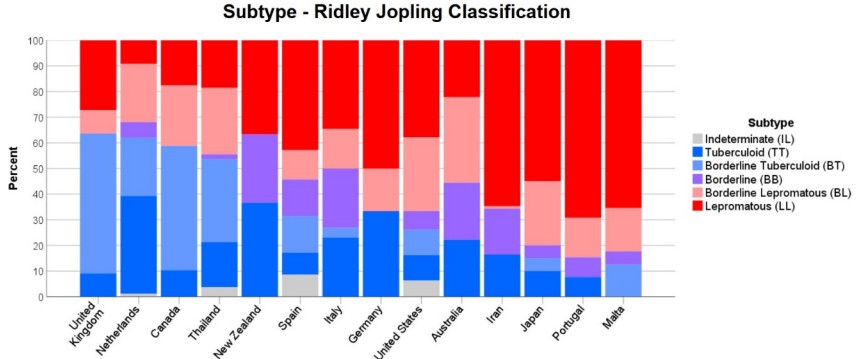

**Fig 5. Leprosy subtype per country using the Ridley Jopling classification system.**

**Table 6. Reported cases with no previous treatment for leprosy confirmed and those suspected to have relapsed.**

| Country | No Previous Treatment | | Suspected Relapse | |
|---|---|---|---|---|
| | **N** | **%** | **N** | **%** |
| Australia | 6 | 100 | 0 | 0 |
| Canada | 117 | 62.9 | 69 | 37.1 |
| Germany | 7 | 87.5 | 1 | 12.5 |
| Iran | 8 | 67.7 | 4 | 33.3 |
| Italy | 6 | 100 | 0 | 0 |
| Malta | 134 | 98.5 | 2 | 1.5 |
| Oman | 77 | 100 | 0 | 0 |
| Portugal | 2 | 16.7 | 10 | 83.3 |
| Spain | 13 | 52.0 | 12 | 48.0 |
| Taiwan (Republic of China) | 68 | 84.0 | 13 | 16.0 |
| United Kingdom | 5 | 50.0 | 5 | 50.0 |
| United States | 53 | 89.8 | 6 | 10.2 |
| Total | **496** | **80.3** | **122** | **19.7** |

was highly endemic in the indigenous population in northern Australia up until the 1960s, resulting in the implementation of widespread BCG vaccination in the Aboriginal population in 1958. The current policy is to vaccinate all Aboriginal neonates living in Aboriginal communities and those born to mothers who have previously been treated for leprosy, with coverage now exceeding 90%. As the incidence of leprosy continued to fall in Australia, contact tracing and surveillance programmes were the primary methods of identifying new cases. From 1971, all newly diagnosed leprosy patients (and previously diagnosed MB patients) were given a three-month course of rifampicin daily in addition to DDS. Clofazimine was later added to existing treatment regimens, with MDT now used as the minimum treatment standard [20].

Case data on 11 leprosy patients diagnosed between 1999 and 2017 was retrieved from eight source publications from Australia [19, 21–27]. The mean case detection delay from five cases in which it was described was 22.8 months. Three cases reported having a family member who was previously diagnosed with leprosy. The number of new recorded cases has plateaued in Australia over the past two decades, from 45 between 2005 and 2009 (11 of them Aboriginal Australians) to 35 between 2016 and 2018 (28 foreign-born) [25, 28, 29]. The characteristics of the 11 cases presented in this study are in line with these recent reports, with a high proportion of MB cases and some local transmission observed in the Aboriginal community.

**Canada.** A majority of the leprosy case data from Canada presented in this study was retrieved from Boggild, *et al.* (2004) who described 184 patients diagnosed from 1979 to 2002 [30]. These patients were diagnosed at the Tropical Disease Unit of the Toronto General Hospital which managed over 80% of reported leprosy cases nationwide during that time. Individual data on an additional two case reports diagnosed in 2006 and 2017 were also included [31, 32]. In total, all but two of the cases described were imported (98.9%), with infection suspected to have originated from leprosy endemic areas outside of Canada. All but one of the cases with information available on treatment received MDT according to WHO guidelines. One of the key findings from the data was the high proportion of cases (37.1%) who received previous treatment for leprosy, indicating a suspected relapse. However, since all of these suspected relapse cases were imported cases originating from endemic countries such as India, the Philippines and Vietnam, it is not known whether they occurred following a full or interrupted course of MDT, DDS monotherapy or another treatment regimen. Interestingly, these suspected relapse cases came from across the Ridley Jopling spectrum of disease classification (7

TT, 23 BT, 12 BL and 16 LL) and not predominantly from subtypes with a high bacillary load. From 21 cases where a potential source was mentioned, 20 were suspected to have come from a family member who had leprosy, while for one Canadian male who's only travel was to Florida, United States, it was reported that the *M. leprae* isolate from his biopsy was identified as the armadillo-associated genotype 3I-2-v1.

The main leprosy control measures implemented in Canada date back to 1906, with the introduction of the "Act Concerning Leprosy", a law administered by medical officers of the Quarantine Service which mandated compulsory confinement of all newly diagnosed leprosy patients. Following successful treatment in the hospital or at home, two bacteriological examinations of smears per year had to be negative for *M. leprae*, with continued surveillance thereafter [33]. In the proceeding decades, treatment regimens were introduced including sulfones, followed by MDT. Canada also introduced BCG vaccination in the first half of the 20th century, but was discontinued as a routine vaccination in the 1970s [34]. From recent WHO leprosy reports and the cases presented by Boggild, *et al.* (2004), it is clear that autochthonous transmission of *M. leprae* is now very limited in Canada. In fact, there were no new cases registered in 2018, even in migrants [29].

**China.**  In this study, we present case characteristics from three studies in Shandong province only; one of the most densely populated provinces located in the eastern part of the People's Republic of China. In 1955, a comprehensive leprosy control programme was introduced with the establishment of the Provincial Skin Disease Control Institute and a network of Skin Disease Control Stations and leprosy hospitals [13]. Since its formation, over 50,000 cases have been registered, which made it one of the most highly endemic provinces in the country up until recently [35]. The prefecture-level city of Weifang in Shandong for example began to keep leprosy under control before the implementation of MDT, reducing the detection rate from 35.2/100,000 in 1956 to 0.7/100,000 in 1985 and further decreasing with MDT (under 0.1/100,000 in 1993) [36]. There are several factors cited which contributed to the decline of leprosy in Shandong: rapid economic development, effective case finding, isolated treatment of patients with MDT (and DDS monotherapy prior to the introduction of MDT) until smear negative, compulsory contact examination and BCG vaccination, which was initiated in the 1970s and reached 90% coverage in new-borns and schoolchildren by 1990 [36–38]. We collected data on 785 leprosy cases diagnosed in Shandong province from 1990 to 2017 [35, 37, 39]. All of the cases reported from these articles were suspected to be autochthonous and the mean case detection delay was 31.7 months. From 778 cases in two of the source publications [35, 37], a total of 137 (17.6%) reported having a family history of leprosy. Other suspected sources included known contact with a leprosy patient, such as a neighbour or friend.

The incidence of leprosy in China as a whole is currently low and continuing to decline, with under 1,000 annual cases reported since 2013 for a population of approximately 1.4 billion [40]. Following the implementation of MDT in 1986, China achieved their goal of elimination (defined as less than one per 100,000) at a national level in 1994 [35]. From an epidemiological summary of cases in China in 2017, 634 newly detected cases of leprosy were reported, with an uneven distribution observed throughout the country. Yunnan province reported the largest number of newly detected leprosy cases (159 cases), followed by Sichuan, Guizhou and Guangdong. There were nine child cases in 2017 and a total of 52 relapse cases reported, 28 of them occurring after MDT [40]. A majority of these cases were MB (91.8%) and the one fifth (20.0%) had grade 2 disability (G2D). One contributing factor to this high proportion of GD2 could be a long average case detection delay across China. Furthermore, Wang, *et al.* (2020) recently reported a very high G2D rate in Shandong of 42.9% [38], which could be partly attributed to the large mean detection delay of 31.7 months observed from case data collected in this study.

**Democratic Republic of Congo.**   In a publication by Tonglet, *et al*. (1990), the decline in leprosy endemicity in Uele, North Eastern Zaire, now the Democratic Republic of Congo (DRC), was examined between 1975 and 1989 [41]. Over this period, the authors showed a drastic reduction in cases under treatment (from 14,445 to 1,050), as well as the number of new cases detected (from 2,963 to 213). The most important causal factors proposed were the distribution of DDS in 1975, given to MB patients as treatment for life and to PB patients for 5–10 years at 300mg/week, which was increased to 600mg/week in 1979. MDT was then introduced as the treatment regimen in 1981 following an increase in DDS resistance. The authors also propose two other factors potentially contributing to the decline in this setting: an improvement in the socioeconomic situation of the population and the introduction of BCG vaccination in 1970. In addition to the reduced new detection rate and improved diagnosis methods, the steep decline in prevalence could be attributed to the successful introduction of rifampicin and the use of combined treatment regimens of limited duration, resulting in many patients being removed from the national registry.

Although this data from Uele illustrates an impressive decline in local leprosy prevalence and case detection in a hyperendemic region, it should be recognised that this is a relatively old study conducted in a very remote area, possibly with many undiagnosed cases remaining. The authors highlight that the previous network of treatment dispensaries in the region was interrupted in the decade prior to the study period, so the large number of new and existing case figures from 1975 onwards probably represent a backlog of cases in a previously under-served population. Moreover, a considerable decline in leprosy prevalence was also observed in many parts of the world during this time period, which has been largely attributed to the introduction of MDT [10]. The DRC is now one of the 23 WHO global priority countries, with 3032 new leprosy cases recorded in 2019 [1]. Given the recent political and economic instability in the country, the ongoing transmission of *M*. *leprae* and other neglected tropical diseases broadly reflects the importance of socioeconomic status as a contributing factor.

**Germany.**   Like many other European countries, leprosy has become a very rare disease in Germany over recent decades. BCG vaccination was introduced in the mid-20th century in East and West Germany, with routine vaccination stopped in 1998 [34]. Between 1982 and 1992, there were 73 new cases reported in former West Germany, all imported from abroad. In 2003, there were only three cases recorded in Germany, two in 2016 and none in 2018 [28, 29, 42]. Given that virtually all new cases are imported from endemic areas, increased migration to Germany may give rise to known risk factors for *M*. *leprae* transmission for asylum seekers with regards to crowding and poor hygiene standards within refugee camps, as well as the low socioeconomic status of many of these individuals. Currently, a new leprosy case in Germany is only notifiable if laboratory confirmed [43].

In this study, we extracted data from six reports with details on eight leprosy patients diagnosed in Germany between 1994 and 2016 [42–47]. There was only one suspected autochthonous case reported, with the remaining 7 (87.5%) imported from countries across Asia. The mean case detection delay for all 8 patients was 22.9 months. All of the cases with treatment information available received MDT, with one suspected relapse reported. There were three cases which reported a suspected source of transmission, one with a family history, another with a previous household contact known to have leprosy, while the third was a rare case of documented patient-to-surgeon transmission of *M*. *leprae*. This originated from an open-muscle biopsy of an LL case presented here, in which the surgeon injured the dorsum of his finger with a scalpel blade. According to the report, the small lesion was immediately disinfected and dressed, but three years later he noticed tenderness of the dorsal branches of the left ulnar nerve without skin involvement. Another two years later, a histopathologic diagnosis of TT leprosy was made [44].

**Iran.**    Data was collected on a total of 207 leprosy cases from two sources, with the date of diagnosis ranging from 1991 to 2009 [48, 49]. The vast majority of cases were suspected to be autochthonous, with only 2 (1.0%) reported cases having migrated from outside of Iran. According to the WHO classification system, most of the cases (83.6%) were MB, while under the Ridley Jopling system the highest proportion were classified as LL (64.7%) followed by BB (17.9%) and TT (16.4%). A skin biopsy was performed on all 207 cases to support the diagnoses, with (81.6%) reporting a positive result. Following confirmation of diagnosis, all cases were treated with WHO recommended MDT regimens for leprosy. One of the source publications reported four patients with a history of disease recurrence and readmission following MDT [49]. From all of the cases reported in both studies, a little over one-third (34.3%) came from rural areas.

According to the WHO, the incidence of leprosy continues to decline in Iran, with only 29 new cases registered in 2018 (28 of those MB), a slight reduction from the 32 new cases reported in 2017 [28, 29]. This decline has been attributed to early case identification and registration, as well as compliance with MDT since its introduction in the 1980s. BCG vaccination at birth was also introduced into the national schedule in 1984 and continues to be administered universally [34]. A majority of the data presented in this study came from 195 cases diagnosed over a 15 year period in the Northwest of Iran [48]. This study noted a positive correlation between age and a shift towards the lepromatous end of the disease spectrum. Interestingly, the second source article from Hamadan in the West of the country reported that four patients out of the 12 presented had a history of recurrence and readmission following MDT (13). Possible reasons stated by the authors were potential drug resistance or incorrect use of drugs by the patients.

**Italy.**    Leprosy incidence has steadily decreased over the past century in Italy and cases are very sporadic, with only 20 new cases recorded in 2016 and 2017, 17 of them foreign-born [28, 50]. The country established the National Leprosy Register in 1923 and all new cases are notified to the Ministry of Health. Compulsory BCG vaccination was introduced for selected groups in 1970 but was never universally adopted throughout Italy [34]. There has been a recent shift towards focusing on migrant populations and the reintroduction of leprosy, as autochthonous cases are now quite rare and more recently thought to have occurred in individuals who spent time aboard in high endemic regions or those who were presumably infected several decades earlier when leprosy was still endemic in Italy [51].

Individual case characteristics were compiled from 27 leprosy patients diagnosed in Italy between 1992 and 2017 described in 10 source publications [51–60]. A large proportion of these cases were male (85.2%), while nine were suspected to be autochthonous versus 15 imported from highly endemic areas. From 26 cases with information on clinical subtype, the majority (73.1%) were MB according to WHO classification with LL the most common subtype (34.6%) under Ridley Jopling, followed by BB (23.1%) and TT (23.1%). There were 10 skin biopsy results presented, nine of them positive. There were also two cases who reported having contact with a family member affected by leprosy.

**Japan.**    Leprosy incidence in Japan has continued to decline throughout the past century and autochthonous transmission has now thought to have stopped throughout most of the country. In 2016 and 2017, only 5 cases of leprosy were recorded and all but one of those were foreign-born [28, 50]. In 1907, hospitalisation of leprosy patients was the main control strategy in Japan and was made mandatory under law in 1931. The Leprosy Prevention Law was further revised in 1953, leading to segregation of patients in a national leprosarium until its abolition in 1996. As was the case in many other countries, patients in Japan began to be treated effectively with MDT in the 1980s leading to a gradual decline in new cases [61, 62]. BCG vaccination for children was first introduced in Japan in the 1940s and continues to be administered, with coverage almost universal in the population [34].

In this study, a total of 20 case reports were compiled from nine publications on Japanese patients diagnosed with leprosy between 1990 and 2017 [63–71]. The mean age in Japan was high compared to most other countries included in the study (61.4 years). There was also a long average case detection delay of 76.3 months, although information was only available from six patients and highly skewed by a 73-year-old male BL leprosy patient with a 33-year history of symptoms prior to diagnosis. This same patient was also suspected to be the source of spousal transmission, in which his wife later developed a BT leprosy after living together for many decades [70]. The only instances of direct family history of leprosy came from two sisters from Brazil who were working in Japan [66]. A majority of the cases were MB (85.0%) and on the lepromatous spectrum of disease according to Ridley Jopling classification (55% LL and 25% BL overall), with 10 out 11 biopsies performed returning a positive result. From 11 cases with information available on treatment, only 2 (18.2%) received MDT, while the rest were treated with DDS monotherapy or other regimens using minocycline and ofloxacin.

**Libya.**   A descriptive study was identified which presented case characteristics of 54 leprosy patients at a clinic in Benghazi, Libya over a four year period, from 1994 to 1998 [72]. The study described a relatively young patient population with a mean age of 35.0 years and mostly male patients (88.9%). There were 47 (87.0%) individuals of Libyan decent which were suspected autochthonous cases. All of the cases received MDT according to WHO treatment guidelines, although more than half were described as having poor compliance to treatment. The authors also described a consistent decline in the number of registered leprosy cases on MDT over the study period, from 18 in 1994 to four in 1998. In terms of overall disability rate, 16.6% of patients included in the study had grade 1 disability (G1D) or G2D. From 47 cases with information on case detection delay, 51.1% reported a delay of under one year while 34.0% experienced a long delay of 3–5 years. There were 18 (38.3%) patients included in this study who had a history of contact with a known leprosy case within their family.

As with many other countries, the decline in leprosy in Libya coincided with rapid economic growth, particularly in the early 1970s. The subsequent elimination of leprosy as a public health problem was mainly attributed to health education, BCG vaccination of neonates, continued socioeconomic development and the introduction of MDT. In fact, the national programme in Libya incorporated the use of rifampicin or clofazimine, as well as lifelong DDS in the treatment of LL cases long before the MDT declaration by the WHO [72]. Despite the political instability over recent years, only 2 cases of leprosy were reported in Libya in 2019 [1]. The number of reported leprosy cases in the study area of Benghazi has reduced steadily over previous decades, from 209 between 1983–93 to 47 in 1994–98 [72]. However, given the large gender disparity observed in this dataset, it is likely that many female cases remained undiagnosed in the region.

**Malta.**   The country of Malta is made up of three islands and leprosy has been present there since at least 1630. The Malta Leprosy Eradication Project (MLEP) programme began in June 1972 and was formally concluded on 31 December 1999, involving retreatment of all known cases in Malta as of 1972 and all new cases thereafter with a fixed regimen of rifampicin and isoprodian (a combination of DDS, prothionamide and isoniazid). It was the first leprosy eradication programme carried out with this particular treatment combination. There were no new child cases found in Malta after the first year that treatment was initiated. By the year 2000, 260 cases had been treated as part of MLEP, including 59 new patients (6 TT, 4 BT, 1 BB, 15 BL and 33 LL). Following widespread BCG vaccination in the 1950s, leprosy incidence was already beginning to decline prior to the start of MLEP and continued to steadily decrease over the 30 year time period of the programme. Leprosy was declared eradicated altogether from Malta in 2001, shortly after MLEP concluded [73, 74].

A summary of 136 individuals treated for leprosy and followed up as part of the MLEP programme from 1971 to 2000 was described by Jacobson, R. R., & Gatt, P. (2008) [73]. Over the

entire duration of the programme, there were 2 (1.5%) instances of treatment relapse recorded. The cases described were mainly MB (87.5%), while according to Ridley Jopling classification, 89 (65.4%) were LL, 23 (16.9%) were BL, 7 (5.1%) were BB. The remaining 17 (12.5%) PB cases were all classified as BT, with no TT cases reported. As part of the MLEP programme, the treatment regimen described was given for five months to those with inactive disease at the start of implementation for a duration of around 24 months (range of 5 to 88 months), while BI positive cases received a longer treatment cycle. From the individuals described in this review, 62 (45.6%) were skin smear positive.

**Morocco.**   In 1991, Morocco achieved the goal of leprosy elimination as a public health problem, defined as less than one case per 10,000 habitants, reflecting a global trend in leprosy decline. Leprosy control measures are coordinated by the National Leprosy Control Program with a focus on early case detection and treatment, contact surveillance and access to health services for rural communities. BCG vaccination is part of the national schedule in Morocco and given at birth [34]. To continue working towards eradication of leprosy, single dose rifampicin (SDR) chemoprophylaxis was introduced for household contacts of registered and newly diagnosed leprosy patients in 2012. As part of the SDR control programme, 5201 household contacts were registered in Morocco, with 4019 (77.3%) contacts examined during investigations, of whom 3704 (92.2%) received SDR.

An overview of 801 leprosy cases diagnosed in Morocco between 2000 and 2017 was detailed by Khoudri, *et al.* (2018), with a focus on the decline observed since the introduction of SDR in 2012 [75]. There were 48 children diagnosed during this period and 78 patients presenting with G2D. A majority of cases (72.4%) resided in rural areas of the country. The number of new cases detected has continued to decrease over the past two decades, from 61 in 2000 to just 13 in 2017 (2 foreign-born). The authors also conducted a time series analysis to further investigate the decline since the introduction of SDR, in which they observed a significant reduction in case detection of 16% per year. This reduction was more pronounced than the previous period of time period of 2000 to 2012, suggesting that the SDR control programme in Morocco was a successful example of a relatively low endemic country taking an innovative approach towards achieving zero leprosy.

**The Netherlands.**   Leprosy is now a very rare disease in the Netherlands, with only 8 new cases reported in 2016 and 2017, all of them foreign-born [28, 50]. In fact, the last autochthonous case in the Netherlands is thought to have had disease onset in 1958 [76]. BCG vaccination is only recommended for specific groups in the Netherlands [34]. A retrospective study presenting the epidemiology of all new leprosy patients diagnosed between 1970 and 1991 was described by Post, *et al.* (1994) [77]. During this time period, there were 622 new leprosy patients recorded in the Netherlands. A majority of these cases were Surinamese (73.3%) where leprosy was once hyperendemic. Unlike many other contexts of leprosy decline, a majority of cases were PB (61.6%). Using the Ridley Jopling classification system, the most common subtype overall was TT (38.2%), followed by BT (22.7%) and BL (22.7%), while only 9.0% were classified as LL.

**New Zealand.**   For many decades now, leprosy in New Zealand has been rare with a majority of cases originating from endemic countries. In the three decades prior to 1980, there were 89 leprosy cases recorded with nearly all of them coming from overseas. BCG vaccination was first introduced in the mid-20th century but was gradually phased out in the proceeding decades [34]. The incidence of leprosy has remained low, with 38 confirmed or probable leprosy cases notified from 2004 to 2013 and just three new cases in 2017. Over the past decade, a majority of cases were reported from the Auckland region and only one case in the South Island. Yu, *et al.* (2015) described these 38 leprosy cases.

All of these cases were imported (mostly from Asia and other Pacific Islands) except for the one case, which was unknown. There were five children aged under 15 years. From 30 cases with a confirmed clinical subtype, most were MB (63.3%) according to the WHO classification system, while TT and LL both made up 36.7% of these cases using the Ridley Jopling system. The remaining 26.7% were classified as BB. There were 9 (23.7%) individuals who reported having contact with a leprosy case within their family. From 19 cases with information on case detection delay, 47.4% reported a delay of less than 1 year, while 21.1% reported a long delay of over 5 years [78]. It is clear from this data that essentially all cases in New Zealand are imported. This absence of autochthonous cases and the consistently low number of new cases reported indicates that transmission of *M. leprae* within the country appears to be very limited, despite the presence of several imported cases on the lepromatous spectrum.

**Norway.**   One of the most well documented declines in leprosy at a national level comes from Norway. The Norwegian leprosy register was the first national disease register in the world and actually preceded the identification of the *M. leprae* bacillus as the causative agent of leprosy by Gerhard Armauer Hansen in 1873 [76]. Between 1850 and 1920, the disappearance of leprosy was observed in Norway despite the absence of effective treatment and prevention options such as DDS monotherapy, MDT or BCG vaccination. The number of new cases detected in Norway declined from 2833 between 1856 and 1860 (including 1796 recorded in the first year of the registry alone) to just 72 between 1911 and 1920. A shift in age of new cases was also observed, with over 60-year-olds accounting for 14% of new cases between 1856 and 1860, and 30% of new cases between 1901 and 1920. Accordingly, the incidence rate in children also declined substantially over this time period, indicating reduced infection within the household. There was also a higher proportion of lepromatous cases (53.8%) observed over the study period. The sex ratio remained consistent over time, with males accounting for 57.1% of new cases diagnosed between 1851 and 1860 compared to 59.5% between 1911 and 1920. The mean case detection delay also declined drastically over time, from 7.8 years in 2289 patients diagnosed before the registry was established in 1856 to 3.2 years in 160 patients diagnosed between 1901 and 1920 [11, 79].

An important factor thought to have contributed to the decline, particularly at the beginning, was physical isolation. Following the introduction of legislation in 1877 and 1885, registered leprosy patients were to be isolated under law either in a hospital or in separate rooms in their home [79]. It was proposed by Irgens, *et al.* (1980) that geographical distribution may have played a role in the occurrence of leprosy in Norway, particularly rural areas and coastal regions with high air humidity which could promote growth of *M. leprae* in the environment. In addition to physical isolation, other important factors thought to have contributed to the decline from high to low endemicity over this period are improved socioeconomic status, reduced malnutrition and selective emigration of high-risk groups [11].

**Oman.**   Oman's National Leprosy Program was established in 1981 with the aim of reducing the burden of leprosy throughout the country. They collaborate with the ministry of health, primary, secondary and tertiary healthcare services, as well as the national programme for tuberculosis. Household contacts of leprosy patients are screened and followed up annually for up to five years. The programme carries out control activities, such as surveillance, laboratory testing of suspected cases and more recently offering pre- and post-test counselling services. BCG vaccination is also administered nationally at birth. Oman achieved the WHO target of eliminating leprosy as public health problem in 1996 and has since continued efforts to achieve zero leprosy, with only one new case reported since 2018 [1, 29].

The case characteristics for 77 leprosy patients diagnosed between 2000 and 2015 were described by Al Awaidy (2017) [80]. Oman has many foreign-born workers and almost a third (31.2%) of the leprosy cases reported here came from outside of the country, mostly from

India, Bangladesh and Pakistan. There were six child cases diagnosed during the study period, with the last recorded in 2014. The study also reported three (4.0%) patients with G2D. Most cases were MB (77.9%), though almost all of the cases (97.4%) were skin smear or biopsy positive. MDT completion was 100% with no relapse cases notified.

**Portugal.**   Portugal experienced a steady decline in leprosy incidence during the second half of the 20th century which coincided with socioeconomic improvement. During the 1950s, there were more than 100 new cases reported per year, which reduced to approximately 20 per year in the 1980s and around single digits in the 1990s [12, 76]. In 2019, there were six new cases reported in Portugal, all of them foreign-born [1]. In a study by Goncalves, *et al.* (2020), 15 individual leprosy cases in Northern Portugal were described in detail out of 38 presented overall [76]. All of the non-Portuguese imported cases were from Brazil, the country with the second highest incidence of leprosy globally. There was limited information available on the actual treatment regimen, but strikingly, 10 out of 12 cases (83.3%) had a suspected relapse from previous therapy with instances of multiple relapses spanning several decades. There were a range of suspected sources of transmission recorded in the case reports, including six cases reporting a family history of leprosy and several individuals who were likely infected during visits to endemic countries.

Compulsory notification of leprosy and other infectious diseases was introduced in Portugal in 1901, followed by additional measures such as mandatory hospitalisation, establishing a national registry and BCG vaccination at birth [12, 81]. The distribution pattern of new cases by age at onset appears to have changed considerably in Portugal, as demonstrated by Irgens, *et al.* (1990) where the median age at onset declined from 30.0 years in 1946–50 to 43.5 years in 1976–80. Given the long incubation time of leprosy and the fact that many recently diagnosed patients are generally older individuals living in rural areas with a high proportion of treatment relapse, it is likely that any local transmission occurred several decades earlier prior to the initial diagnosis [76].

**Saudi Arabia.**   Although leprosy has been a notifiable disease in Saudi Arabia since 1963, the most substantial decline in incidence followed the introduction of a case reporting system for leprosy and other infectious diseases by the Ministry of Health in 1985 [82, 83]. BCG vaccination was introduced nationally in 1968 and coverage is now almost universal [34]. In conjunction with an improved nationwide primary health services network, Saudi Arabia achieved the WHO elimination target of one case per 10,000 population. However, the fact that new cases were still being reported within the country suggested that transmission of *M. leprae* was ongoing, leading to an enhanced surveillance system being established in 2003 to be used together with active case finding activities [83]. With the introduction of these control measures, the number of new cases in Saudi Arabia gradually declined over time from 262 in 1986, 169 in 1989 and 42 in 2003 [82, 83]. In the most recent WHO global leprosy update, there were 17 new cases recorded in 2019, all of them MB [1].

A report of 242 leprosy cases diagnosed between 2003 and 2012 was presented by Assiri, *et al.* (2014) [83]. There were a total of four children under the age of 15. The report also contained data on disability grade, which described 38 (15.7%) individuals presenting with G1D and 22 (9.1%) with G2D. Overall, there were 103 (42.6%) Saudi patients suspected to have been infected within the country and 139 (57.4%) non-Saudi cases from mostly South Asian descent, such as Indian, Nepalese and Bangladeshi. In order to further reduce the risk of transmission within the country, the public health authority in Saudi Arabia has a policy to treat all non-Saudi cases for only one month, after which time they must return to their respective countries. Since infected non-Saudis may be aware of this policy, there could be many undiagnosed foreign-born cases who do not seek treatment for fear of repatriation.

**South Korea.** An epidemiological report by Lee, *et al.* (2015) presented data on new leprosy cases diagnosed in the South Korea between 1989 and 2013 [84]. The number of new cases declined from 62 in 1989 to only four in 2019 (all four foreign-born). There are several factors which may have contributed to this decline: improved socioeconomic status, nutrition and hygiene throughout the population. BCG vaccination was first implemented in the South Korea in 1954, with around 70% coverage reported in 1990. MDT was also introduced in 1982, with a national standard for treatment completion established three years later. All MB cases reported in the country receive MDT until smear negative, while PB cases are treated for two years. A similar pattern was also observed in this report which has been previously documented during the decline of leprosy in many other countries, namely an increased proportion of cases with a family history, as well as a shift towards MB subtype and older age groups. In fact, there were no new child cases reported in the most recent time period between 2009 and 2013. Moreover, a majority of new leprosy cases are imported, indicating that autochthonous transmission of *M. leprae* has substantially declined within the country.

**Spain.** At the beginning of the 20th century, Spain had several leprosy hospitals for the treatment and isolation of patients. The national leprosy control programme was only established in the 1940s after the civil war, which introduced contact tracing and active case finding activities, leading to an increase in new cases shortly thereafter [85]. The incidence of leprosy has steadily declined in Spain since the 1950s, with a majority of cases now imported from endemic countries in Latin America such as Brazil [85, 86]. In 2017 and 2018, there were 16 new cases of leprosy recorded, 12 (75.0%) of them foreign-born [28, 29]. Sulfones were first introduced as a treatment for leprosy from 1945, which was then replaced by MDT in 1981 [85]. Spain also introduced universal BCG vaccination in the 1960s, which may have contributed to the decline, although the programme was stopped in 1981 [81].

Data was collected from seven source publications on 97 leprosy cases diagnosed across Spain from 1989 to 2018 [85, 87–92]. Overall, a majority of cases were MB (71.4%), while from 35 cases that were classified according to Ridley Jopling, the largest proportion were LL (42.9%). The average case detection delay from 21 cases in which it was reported was 43.0 months. Most of the 28 cases with treatment information available received MDT (67.9%), although some other regimens were also used, including those with minocycline or ofloxacin. There were 14 cases who reported have a family member with a history of leprosy and another three who had a non-family contact with leprosy as the suspected source of transmission. Norman, *et al.* (2016) presented data on 25 leprosy cases diagnosed in Madrid (10 autochthonous and 15 imported) in which 12 (48.0%) cases from across both groups had been previously diagnosed and treated for leprosy. These instances of relapse mainly occurred following DDS monotherapy, although there was also a case with thalidomide and a few instances of MDT, suspected to have been incomplete or given at a suboptimal dose [90].

**Taiwan (Republic of China).** In the early 20th century, a comprehensive leprosy control programme was established in Taiwan, including sanatoriums to quarantine and isolate patients. In the 1950s, a BCG vaccination programme was introduced with DDS used as the primary treatment option, later replaced by MDT in 1983. The official national leprosy control programme of Taiwan was implemented in 1962, which coordinates case detection, treatment and education activities throughout the country [93]. BCG vaccination was also expanded to include new-borns and infants in 1965 [34]. The incidence of leprosy peaked in 1962 with 379 new cases reported, which declined to single digits per year in the 1990s [93]. In Huang & Jou (2014), a summary of 81 leprosy cases diagnosed in Taiwan between 2002 and 2011 was presented [94]. There were 44 (54.3%) imported cases from other endemic areas in Asia, mainly Indonesia (39.5%). None of the cases diagnosed during this period were under the age of 19. Interestingly, 13 (16.0%) cases relapsed from previous treatment, all of them autochthonous.

As part of the national laboratory-based surveillance for *M. leprae* conducted by the Taiwan CDC, polymerase chain reaction (PCR) analysis was conducted on 13 samples. All 13 were found to be susceptible to rifampicin or fluoroquinolone (mutations in the *rpoB* and gyrA genes respectively*)* while two cases were found to be resistant to DDS (*folP* gene).

**Thailand.**   The leprosy decline in Thailand presents an interesting case study as it occurred more recently than many of the examples presented here, such as those in the European region. Specialised leprosy control programmes were expanded throughout the country in the 1970s, covering both low and hyperendemic regions [95, 96]. Around the same time, neonatal BCG vaccination was also introduced in Thailand and continues to be part of the national schedule [34]. DDS monotherapy was the main treatment method during this time with clofazimine added for all new smear positive cases. This led to a decrease in the number of new cases, diagnosis delay and disability percentage, as well as an increase in the average age of patients and a higher proportion of PB cases [97]. By far the most effective reduction was seen in the 10 years following the introduction of MDT in 1984, with the number of registered cases dropping from 44,406 in 1984 to just 4,878 cases in 1994 [96]. Over the past two decades, the incidence of leprosy has continued to gradually decline throughout Thailand, with 138 new cases recorded in 2019, 60.9% of those MB, with 18 reporting G2D and three child cases overall. Foreign-born cases only represented 13.8% of these new cases and there were 13 instances of relapse [1].

Suchonwanit, *et al.* (2015) presented a retrospective study of 108 leprosy reactions in the Thai population over a 20 year period, from 1995 to 2015 [98]. Although this study was conducted in Bangkok, only a little over half (54.6%) of the patients lived in the capital city. The subtype recorded was quite evenly distributed, with 58 (53.7%) MB cases, while according to Ridley Jopling, the highest proportion were BT (32.4%), followed by BL (26.0%), LL (18.5%), TT (17.6%) and BB (1.9%). The remaining 4 (3.7%) cases were described as indeterminant leprosy (IL). There were 59 smear positive patients with a mean BI of 3.4. All patients described in this study received MDT according to WHO guidelines for treatment of PB and MB cases. The source of transmission was suspected in 7 (6.5%) cases, all of whom had a history of contact with somebody affected by leprosy.

**United Kingdom.**   Leprosy first became a notifiable disease in England and Wales in 1951 and has since become a very rare disease. BCG vaccination of children was introduced in 1953 and subsequently discontinued in 2005, with vaccination now only recommended for special groups [34]. Between 1953 and 1962, there were 356 new cases notified, falling to 139 between 2003 and 2012 [99]. Over the past four years, only 18 cases were reported in the United Kingdom, all of them foreign-born [1, 28, 29, 50]. In 1997, Public Health England revised the Memorandum on Leprosy in order to improve case management and notification, while also providing guidance on diagnosis and treatment of leprosy patients [99]. Here we collected data from six individual leprosy case reports and series in the United Kingdom over a an extended period, from 1977 to 2014 [100–105]. All 11 cases reported a skin biopsy result, 9 (81.8%) of them positive. The mean case detection delay was 6.7 months from six cases with information available. Waters & Ridley (1990) described four cases of tuberculoid relapse in lepromatous leprosy after previously receiving sulfone treatment, with DDS resistance proven in one case [101]. One other report described an LL patient who relapsed, initially diagnosed and treated in Nigeria 18 months earlier [105].

Interestingly, there was one 68-year-old Caucasian man diagnosed with LL in 1988, 11 months after he returned to England after spending the previous 40 years in the tropics. Following this, the cellular and humoral response to *M. leprae* was studied in a group of contacts who had been in contact with the index case for over a year, with two young adults found to have raised antibody concentrations. It was deemed that subclinical infection of *M. leprae* may

have occurred in these two contacts and they were subsequently given 6 months chemoprophylaxis with rifampicin [104]. Another important finding came from Avanzi, *et al.* (2016) who detected *M. lepromatosis* in red squirrels from England, Ireland, and Scotland and *M. leprae* in squirrels in Brownsea Island, England. This unexpected discovery was the first report of a non-human animal reservoir for leprosy outside of armadillos in the Americas [106]. However, no evidence was found for the presence of these mycobacteria in red squirrels from the mainland [5].

**United States.** The contemporary history of leprosy control measures in the United States dates back to the 19th century, when Hawaii and Louisiana introduced isolation measures for leprosy patients in 1865 and 1894 respectively. Following this, a series of national legislative acts were passed mandating quarantine and treatment in a leprosarium. The American Public Health Association advised against isolating people affected by leprosy in 1945, which was followed by a gradual expansion of basic human rights for patients in the proceeding decades [107]. Sulfones were introduced as the preferred treatment option in the 1940s, although the United States never implemented a universal BCG vaccination programme [107, 108]. The incidence of leprosy in the United States had been declining since the 1980s, with 102 new cases reported in 1998 [109]. There has been an increasing trend over the past decade however, with 178 new cases reported in 2015, 168 in 2016 and 185 in 2018 [29, 50, 110]. Most new cases were reported in the Southern states where nine-banded armadillos are present as a potential source of transmission, including Texas, Louisiana and Florida, although cases have also been reported in California, New York and Hawaii mainly in immigrant populations [109–112]. It has been recently reported that armadillos are likely involved in up to two-thirds of all new leprosy cases recorded each year in the United States [113].

Data was collected from 38 source publications, with information on 303 cases diagnosed across the United States between 1982 and 2018 [3, 109–146]. There were 214 cases suspected to be imported from outside of the United States, including 177 from Micronesia and the Marshall Islands between 1990 and 2009 as described by Woodall, *et al.* (2011). There were 59 cases with biopsy results reported, 55 (93.2%) of them positive. The mean detection delay observed was relatively low at 16.5 months from a total of 43 cases. There were 248 cases in our dataset which described a suspected source of transmission, with only six of these reporting a family history. The remaining cases came from contact with a known leprosy case, travel to an endemic region or contact with armadillos. In total, we found 46 cases diagnosed over this time period who reported a history of contact with armadillos, mostly handling with some cases of consumption of armadillo meat [110, 112, 113, 115, 119, 126, 130, 132, 135, 136, 144, 146].

Where treatment information was available, most cases received MDT (79.2%). There were six instances of suspected relapse, of which three were LL, two were BL and the other was BB [119, 133, 144, 147]. Three of these cases were treated with MDT (one with poor compliance), two with minocycline containing regimens and the other relapse occurred following treatment with thalidomide and prednisone (although prednisone was not taken initially due to a miscommunication). One of the relapsed BL patients originally treated with MDT was a confirmed case of multidrug resistant leprosy, with molecular genotyping revealing mutations characteristic of DDS and rifampicin resistance [142]. A retrospective chart review of 151 leprosy patients treated with MDT in the United States between 1988 and 1997 and followed up for 10–15 years was described by Dasco *et al.* (2011) [148]. Contrary to the individual reports of disease recurrence presented here, only one case of relapse was reported in the study; an MB patient who underwent two drug therapy (DDS and rifampicin), indicating that relapse following a full course of MDT is uncommon in this context.

**Vietnam.** Prior to the formation of the National Leprosy Control Programme in 1982, leprosy was a major public health concern in Vietnam. They coordinate basic leprosy services,

including technical guidance, referral of suspected cases and treatment follow up for confirmed cases [149]. One year after the formation of the programme in 1983, MDT was implemented, followed by BCG vaccination for neonates in 1985 [34, 149]. As was the case with neighbouring Thailand, a significant decrease in prevalence was observed in the 10 years following the introduction of MDT. As a result of this and other long-term control efforts, Vietnam achieved the WHO elimination target at the national level in 1995. Despite these encouraging figures, detection of new cases actually increased slightly over this time, from 2021 in 1983 to 2591 in 1995. Over the past two decades however, the incidence of leprosy has declined remarkably, from 1477 in 2000 to just 77 in 2018.

A comprehensive epidemiological report of leprosy in Vietnam described in Khang, T. H., & Thao, N. M. (2019), with 96 cases registered in 2018 presented here [149]. There was a sharp decline in the number of new child cases reported in Vietnam over the past two decades, from 105 in 2000, to just 14 in 2010. There were no child cases reported in 2018, suggesting that transmission of *M. leprae* has been interrupted. A vast majority (92.7%) of cases were MB with a high proportion of G2D (18.8%). The high number of MB cases continues an ongoing trend over time, from 40.1% of cases in 1983, to 62.0% in 2000 and now over 90% in 2018. This finding could in part be explained by a change in the criteria for classification of leprosy over time, but is also representative of the shift towards the MB spectrum observed in many other contexts during periods of declining incidence. All of the patients described in this study were treated with MDT according to WHO guidelines. In 2018, there were five patients who had reportedly relapsed from previous treatment [29].

## Discussion

In this systematic review, we aimed to evaluate the case characteristics during the declining stages of leprosy incidence, to identify the possible remaining sources of transmission in low endemic settings and to relate these findings to the different leprosy control measures implemented. Together with socioeconomic improvement over time, there were some notable strategies shared by many of the countries who achieved a substantial reduction in incidence over recent decades. These included BCG vaccination, active case finding, adherence to MDT and continued surveillance following treatment. Despite a high proportion of the multibacillary forms of the disease and the presence of persistent cases of suspected relapse, we found that the number of new cases reported remained low. This evidence suggests that such cases do not represent a considerable source of *M. leprae* transmission in low endemic areas.

All countries included in the quantitative component of this study are now considered low endemic with less than one new leprosy case detected per 100,000 population. Many of the new cases recorded in these settings were imported from other areas of the world. This was particularly true in Canada, New Zealand, South Korea, United States and much of Europe. It can be difficult to determine the exact history of non-autochthonous cases, as information on previous diagnosis, treatment and contact with potential sources of transmission in their country of origin is often unclear. Patients who report travelling or living aboard also tend to be classified as non-autochthonous, although it is not possible to determine where they acquired their infection with absolute certainty. Nevertheless, there is no evidence to suggest that an increase in foreign-born leprosy cases arriving from high endemic areas contribute to a noticeable rise in local transmission.

Previous studies have reported that the characteristics of cases shift towards an older population and predominantly MB subtype during the declining stages of leprosy incidence [11–13]. The same shift was also observed in several low endemic countries presented in our study. A striking example of this was observed in Vietnam, which reported no new child cases in

2018. This impressive decline in child leprosy rate, along with a steep decline in overall incidence, suggests that the national control measures were successful at interrupting transmission throughout the country. A key finding from investigating the case characteristics in these countries was the high proportion of MB cases under the WHO classification and those on the lepromatous spectrum where Ridley Jopling was used; subtypes often confirmed by the presence of bacilli in a skin biopsy. Nevertheless, recent reports did not indicate that such cases led to a rise in new (secondary) cases. For example, in 77 leprosy patients diagnosed over a 15-year period in Oman, almost all were skin smear or biopsy positive and most were classified as MB. However, only one new case has been reported in Oman since 2018. The authors attributed much of this success to 100% MDT compliance and an emphasis on continued surveillance following treatment completion.

Achieving a reduction in case detection delay is becoming an important area of focus in leprosy research since decreasing the time to diagnosis can help prevent disability and shortens the time of possible transmission to others. The average case detection delay in our dataset was 31.4 months, with many individuals experiencing symptoms for long periods of time before being correctly diagnosed. When treated, most cases received MDT according to WHO treatment guidelines for PB and MB cases. Where information was available, there were many cases of disease recurrence following previous treatment, although it was difficult to match all of these cases to clinical subtype and treatment regimen. However, there were several individual reports of older patients with an original diagnosis dating back many decades in which one or multiple relapses occurred following DDS monotherapy. We also found instances of relapse following MDT, but this appears to be rare in the absence of confirmed drug resistance and when treatment compliance was achieved [148].

Regarding possible remaining sources of transmission in these low endemic settings, the most commonly reported were family history, previous contact with a person known to have leprosy and armadillo exposure in North America. There is evidence of a genetic component to an individual's susceptibility to infection, likely resulting from an inability to adequately control the infection through cell-mediated immunity after exposure [15, 18]. Despite the uncertainty regarding exact mode of transmission of *M. leprae*, it has been well documented that prolonged exposure with somebody living with the disease, either living under the same household or within the community, increases the risk of infection [10, 150]. Where no obvious source can be found in these low endemic settings, many cases were likely infected multiple years earlier and diagnosed following very long incubation times.

Aside from *M. leprae* transmission from infected humans, it has also been suggested that environmental sources could also be an important reservoir. In a recent study by Tió-Coma, et al., *M. leprae* DNA was detected in soil from houses of leprosy patients in Bangladesh, armadillos' holes in Suriname and habitats of lepromatous red squirrels in the British Isles [151]. In our study, we also found numerous reports of leprosy cases with a history of contact with armadillos in the United States. A zoonotic transmission pathway from exposure to armadillos has been proposed, with human patients from a previous study in southeastern United States shown to be infected with the same armadillo-associated *M. leprae* genotype [152]. Moreover, high rates of *M. leprae* infection were observed in armadillos in the Brazilian state of Pará, with individuals who frequently consumed armadillo meat showed a significantly higher titres of the *M. leprae*-specific antigen, phenolic glycolipid I (PGL-I) compared with those who did not or ate them less frequently [153].

To better understand the factors which contributed to a decline in leprosy incidence, we also summarised a brief history of control measures implemented in these settings. Among the best documented examples found was the leprosy eradication program in Malta. The programme used a fixed regimen of rifampicin and isoprodian to re-treat all known cases in

Malta at the start of the programme in 1972 and all new cases thereafter, with leprosy officially declared eradicated from the country in 2001. Importantly, no new child cases were found after the first year that treatment was initiated, suggesting that this highly successful programme may not be easily reproducible elsewhere. The reduction in leprosy incidence was also attributed to the introduction of BCG vaccination prior to the start of the programme. Many other countries also introduced the BCG vaccine either universally or for specific groups such as neonates, mainly to control tuberculosis. The cross reactive potential has surely impacted the decline of leprosy, although the level of protection offered by BCG vaccination against *M. leprae* from observations in previous studies have varied considerably [154]. The introduction of chemoprophylaxis in the form of single dose rifampicin in Morocco was another successful strategy. Since it was implemented in 2012, Morocco has moved towards eradication, with a significant reduction in leprosy case detection of 16% per year observed.

Overall, the introduction of MDT as an effective treatment option in the 1980s appears to have had the largest impact on leprosy control. This was first recognised from the reduction in global disease prevalence, but together with active case finding and monitoring adherence, MDT ultimately led to a reduction in the number of new cases reported as well [9]. The effect of rapid improvement in socioeconomic status of a population should not be underestimated either, regardless of geographical location. This transition leads to improved hygiene, living conditions, access to healthcare and minimises potential environmental reservoirs of *M. leprae* [4].

This was the first systematic review to investigate the evidence and sources of ongoing *M. leprae* transmission using case data from across several low endemic countries and compare these findings to previous control measures. It allowed us to gain insights at a country level where the dataset was large enough, but also present characteristics from a variety of global contexts. By conducting a detailed review of case reports, we were able to present multiple cases of confirmed sources of infection. As we extracted data from a range of sources in which data was presented in different ways, we encountered limitations in distinguishing between endemic and imported, as well as matching certain variables to one another, most notably between relapse cases and previous treatment regimens. This made it difficult to perform worthwhile association tests between variables. In our country descriptions we refer to recent WHO reports for indications of a rise in secondary cases. These figures are typically reported by national leprosy programs, whereas the individual cases presented were collected from literature based data and only account for a limited sample of the official data. Additionally, due to the nature of the individual case data collected it was not possible to perform any meaningful time trend analyses or make projections for individual countries. Performing an analysis of pooled case data would also not be appropriate given the broad range of different contexts and timeframes studied. Instead, the aim was to make a single assessment of the leprosy case characteristics over the specified period of incidence decline. Additional variables were also included in our data collection protocol that were, however, too limited to present, such as surveillance period, occupation, suspected reinfection and rural/urban areas. Nevertheless, the importance of access to healthcare and continued surveillance in rural areas was described in some settings, including Iran and Morocco.

The success of various leprosy prevention and control measures, particularly during periods of socioeconomic improvement, is evident from the decline in new cases observed in many parts of the world. However, a better understanding of the ongoing sources of transmission is required to design policies in the WHO global priority countries and to continue working towards zero leprosy. Most of the cases reported during periods of declining incidence were multibacillary with numerous cases of suspected relapse reported. Despite these observations, there was no indication that the cases described here led to a rise in new secondary cases, suggesting that they do not represent a large ongoing source of human-to-human transmission.

## Supporting information

**S1 PRISMA Checklist.**
(DOC)

**S1 List of Peer-reviewed Studies with Case Data.**
(DOCX)

## Acknowledgments

The authors would like to thank the ILEP Technical Commission for their contributions and feedback throughout the project. We would also like to thank the Medical Library at Erasmus MC for their help developing the search strategy for the systematic review.

## Author Contributions

**Conceptualization:** Thomas Hambridge, Annemieke Geluk, Paul Saunderson, Jan Hendrik Richardus.

**Data curation:** Thomas Hambridge, Shri Lak Nanjan Chandran.

**Formal analysis:** Thomas Hambridge.

**Investigation:** Thomas Hambridge, Shri Lak Nanjan Chandran.

**Methodology:** Thomas Hambridge, Annemieke Geluk, Paul Saunderson, Jan Hendrik Richardus.

**Project administration:** Thomas Hambridge.

**Resources:** Thomas Hambridge.

**Supervision:** Jan Hendrik Richardus.

**Validation:** Shri Lak Nanjan Chandran, Jan Hendrik Richardus.

**Writing – original draft:** Thomas Hambridge.

**Writing – review & editing:** Shri Lak Nanjan Chandran, Annemieke Geluk, Paul Saunderson, Jan Hendrik Richardus.

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
