## [Decision Letter · Decision Letter 0]

2 Mar 2021

Dear Dr. Hambridge,

Thank you very much for submitting your manuscript "Mycobacterium leprae transmission characteristics during the declining stages of leprosy incidence: a systematic review" for consideration at PLOS Neglected Tropical Diseases. As with all papers reviewed by the journal, your manuscript was reviewed by members of the editorial board and by several independent reviewers. In light of the reviews (below this email), we would like to invite the resubmission of a significantly-revised version that takes into account the reviewers' comments. 

We cannot make any decision about publication until we have seen the revised manuscript and your response to the reviewers' comments. Your revised manuscript is also likely to be sent to reviewers for further evaluation.

Sincerely,

Linda B Adams

Associate Editor

Godfred Menezes

Deputy Editor

Reviewer's Responses to Questions

**Key Review Criteria Required for Acceptance?**

**Methods**

-Are the objectives of the study clearly articulated with a clear testable hypothesis stated?

-Is the study design appropriate to address the stated objectives?

-Is the population clearly described and appropriate for the hypothesis being tested?

-Is the sample size sufficient to ensure adequate power to address the hypothesis being tested?

-Were correct statistical analysis used to support conclusions?

-Are there concerns about ethical or regulatory requirements being met?

Reviewer #1: No. This is a descriptive study which offers little new insight and little analysis.

Reviewer #2: NA

Reviewer #3: This a systematic review of published literature of leprosy case characteristics in low endemic countries. The authors used standard search methodology and their study inclusion and exclusion criteria seem appropriate.

**Results**

-Does the analysis presented match the analysis plan?

-Are the results clearly and completely presented?

-Are the figures (Tables, Images) of sufficient quality for clarity?

Reviewer #1: Excellent narrative descriptions of leprosy in different countries, but little to no analysis or projection of trends

Reviewer #2: NA

Reviewer #3: Yes

**Conclusions**

-Are the conclusions supported by the data presented?

-Are the limitations of analysis clearly described?

-Do the authors discuss how these data can be helpful to advance our understanding of the topic under study?

-Is public health relevance addressed?

Reviewer #1: The primary conclusion of the study was that 'imported leprosy cases seem to have little impact on the endemic population, and they do not represent a large reservoir for ongoing transmission. This seems obvious from the start and no new analysis is included

Reviewer #2: NA

Reviewer #3: Yes

**Editorial and Data Presentation Modifications?**

Reviewer #1: (No Response)

Reviewer #2: NA

Reviewer #3: (No Response)

**Summary and General Comments**

Reviewer #1: The authors present here a literature review concerning leprosy in low endemic areas with the notion that “As leprosy incidence begins to decline, characteristics of new cases shifts away from those observed in highly endemic areas, revealing potentially important insights into possible ongoing sources of transmission.” They performed an exhaustive literature review covering 22 countries where they suggest 48% of the cases were imported, 64% were multibacillary and 18% had a family history. They conclude that “there was no indication that the [new] cases described here led to a rise in new secondary cases, suggesting that they do not represent a large ongoing source of human-to-human transmission.”

Of course, these general observations are not new, and given the fact the countries reviewed are lowly-endemic, it seems rather obvious that the relative few endemic or imported cases reviewed would not represent a ‘large ongoing source of human-to-human transmission’. While they reiterate that imported cases seem to have little impact on endemic disease rates, they make no attempt to describe what source may sustain low level endemic transmission (other than perhaps some family relationships in up to 18% of cases), or address the classic irony that leprosy appears to have spread around the world by colonization, trade and adventurism; but in modern times introduction from foreign sources seems to have little impact. 

A major difficulty comes in discerning where a case may have acquired their infection. Typically, if a patient reports having traveled or lived abroad at all, programs tend to classify their source to be foreign and also having been acquired abroad. However, the actual voracity of this assumption is actually unknown. Some programs require a diagnosis within 5, 7, or 10 years of the foreign exposure. Others may consider anytime abroad, even if only for short-term travel. The subtlety of discerning an imported from endemic case is not elaborated or considered. How do you know if an infection was acquired abroad? 

The paper includes a large number of figures and tables which are not really utilized analytically. We see that the age, gender, disease-type and ratio of imported:autochthonous cases and relapse varies widely among the countries reviewed. While these figures are a compact way to annotate the individual country narratives, they really don’t reveal anything about how these ‘characteristics’ have changed over time. In addition, these wide variations are likely impacted by the highly variable number of cases reviewed in different countries. Unfortunately, no statistical analysis was used to smooth the ratios or project some trends in the characteristics. Curiously, MDT and BCG vaccinations were highlighted as major contributing factors in achieving control of leprosy in many programs, but again there is no substantive analysis of these ‘characteristics’ between the low-endemic countries or between them and other persistently high endemic areas. 

In recent times the role of non-human reservoirs of M. leprae has gained notoriety in the literature. The importance of these reservoirs in sustaining leprosy is not yet confirmed. The authors do note that armadillos are associated with 64% of the cases in the United States. Does the persistence of leprosy in low endemic areas suggest there might be other non-human reservoirs, especially given the fact that imported cases do not seem to be contributing greatly to ongoing human-to-human transmission? In addition, recent literature suggests armadillos in South America also may be involved in zoonotic transmission. When one major source of infection comes under control, other lessor sources rise in importance. Does persistence of infection in low-endemic areas suggest sources other than human-to-human likely play a role?

Other minor considerations:

Line 1008: incorrectly suggests that humans and armadillos share a single genotype of M. leprae

Line 1009: rather obscure citation. Should cite the original articles.

Line 1043: should include distinguishing endemic/imported as a major limitation.

Reviewer #2: Comments:

The authors performed a systematic review for the literatures of leprosy and summarized the transmission characteristics of mycobacterium leprae during the low incidence of leprosy. The paper has public health implications because it provides different measures for prevention and control of leprosy. 

Major comments:

1) WHO globle leprosy programme will report chracteristics of newly onset leprosy cases in each country. The data is reported by the national leprosy management department. The author should analyze the similarities and differences between the official data and the literature based data. Limitation of the literature based data should also be highlighted. 

2) The country description part is too lengthy. Those data should be better integrated and analyzed.

3) Did the author have the whole list of 105 studies involved in this analysis? It should be put at the supplementary file.

4) Taiwan is an indispensable part of China and has never been a country. The author should revised current statement. 

Minor Comments:

1) Methods: Please clarify the range set of published date of the literature when conducting the literature search.

2) Methods: The authors included “the studies from countries or regions with less than one new leprosy case detected per 100,000 population”, however, the incidence of leprosy in each country varies from year to year. Did the authors use the incidence of the specific year?

3) Results: As for the autochthonous cases reported, the detailed transmission route of these cases should be explained such as relapse, travelling to high endemic areas, or contact with animals like armadillos?

Reviewer #3: In the introduction the authors should mention that leprosy can be caused by M. leprae and M. lepromatosis. 

It may be helpful if the authors mention number of new cases in 2019 (when available) for the countries included in this review. For most countries this is now available in WHO database.

In the discussion the authors may emphasize the importance of access to health care and continued surveillance in rural areas for leprosy control. Although the data for urban vs rural cases were not available for all the reviewed low endemic countries, but countries like Iran, Morocco, Norway & Portugal showed the importance of surveillance in rural areas which for various reasons can harbor a more conducive environment for leprosy transmission than the urban areas.

PLOS authors have the option to publish the peer review history of their article (what does this mean?). If published, this will include your full peer review and any attached files.

Reviewer #1: No

Reviewer #2: No

Reviewer #3: No
---

## [Editor Report · Decision Letter 1]

3 May 2021

Dear Hambridge,

We are pleased to inform you that your manuscript 'Mycobacterium leprae transmission characteristics during the declining stages of leprosy incidence: a systematic review' has been provisionally accepted for publication in PLOS Neglected Tropical Diseases.

Best regards,

Linda B Adams

Associate Editor

Godfred Menezes

Deputy Editor

---

## [Editor Report · Acceptance letter]

21 May 2021

Dear Mr. Hambridge,

We are delighted to inform you that your manuscript, "*Mycobacterium leprae * transmission characteristics during the declining stages of leprosy incidence: a systematic review," has been formally accepted for publication in PLOS Neglected Tropical Diseases.

Best regards,

Shaden Kamhawi

co-Editor-in-Chief

Paul Brindley

co-Editor-in-Chief
